# Evaluation of the Effect of Adipokinetic Hormone/Corazonin-Related Peptide (ACP) on Ovarian Development in the Mud Crab, *Scylla paramamosain*

**DOI:** 10.3390/ani14243706

**Published:** 2024-12-23

**Authors:** Wenting Tan, Yiwei Tang, Fang Liu, Li Lu, An Liu, Haihui Ye

**Affiliations:** State Key Laboratory of Mariculture Breeding, Fisheries College, Jimei University, Xiamen 361021, China; 202211710037@jmu.edu.cn (W.T.); 202111908007@jmu.edu.cn (Y.T.); liufang@jmu.edu.cn (F.L.); 202114908002@jmu.edu.cn (L.L.)

**Keywords:** GnRH-like, adipokinetic hormone/corazonin-related peptide, ovarian development, reproduction, mud crab

## Abstract

Gonadotropin-releasing hormone (GnRH) is an important regulator in the process of reproduction in vertebrates through the hypothalamic-pituitary-gonadal (HPG) axis. Adipokinetic hormone/corazonin-related peptide (ACP) is a GnRH structurally related peptide that is extensively present in arthropods. In insects, ACP is involved in lipid metabolism, but its function in crustaceans remains unclear. This study identified ACP and its putative receptor in the mud crab *Scylla paramamosain* and further evaluated its possible role in ovarian development. Our results showed that ACP was mainly expressed in the follicle cells and showed that it played a stimulatory role in ovarian development. ACP signaling enhances *vitellogenin* (*Vg*) expression in the hepatopancreas, which is likely mediated by estradiol, and promotes vitellogenin uptake by oocytes by inducing *Vg receptor* expression. These findings provide new insights into the role of ACP signaling and hormonal regulation on reproduction in economically important crustaceans.

## 1. Introduction

Neuropeptides play important roles in regulating various physiological processes in animals, including growth, metabolism, development, and reproduction [1,2]. In vertebrates, gonadotropin-releasing hormone (GnRH), a hypothalamus-derived neuropeptide, plays a pivotal role in regulating reproduction via the hypothalamic–pituitary-gonadal (HPG) axis [3,4]. Interestingly, although the lack of analogous glands to the hypothalamus, GnRH-like neuropeptides have been widely identified in invertebrates, including octopus *Octopus vulgaris* [5], abalone *Haliotis asinina* [6], sea hare *Aplysia californica* [7], and sea cucumber *Holothuria scabra* [8]. Additionally, the involvement of GnRH-like peptides in reproductive processes was also demonstrated. It reported that *Oct*-GnRH stimulated sex steroids hormone release in the octopus *O. vulgaris* [9], GnRH peptide induced spawning of gametes from the mature ascidian *Ciona intestinalis* [10], and promoted oocyte proliferation in the abalone *Haliotis discus hannai* [11].

In arthropods, adipokinetic hormone (AKH), corazonin (Crz), and adipokinetic hormone/corazonin-related peptide (ACP) are three neuropeptides structurally similar to the mammalian GnRH [12]. AKH, which is characterized by the core sequence pQXTFSXXWXXamide (X represents variable amino acids), is only found in insects [13]. It has been established that AKH was involved in the mobilization of energy reserved in the fat body and affected the nutritional balance during diapause and oogenesis [14,15]. Crz is a conserved peptide in arthropods, which is 11-aa long and identified by a conserved pQTFQYSRGWTNamide [16]. It reported that Crz was implicated in the regulation of the cardiovascular system, such as accelerating the heartbeat [17,18,19]. Recently, it revealed that Crz was also involved in the regulation of ovarian development and ecdysteroid biosynthesis [20]. ACP is the intermediate of AKH and Crz in structure, which is characterized by a conserved sequence of pQxTFSxxWamide [12]. However, the function of ACP signaling is poorly understood [21]. Past studies indicated that ACP had no significant effect on lipid levels or heart rate [22]. However, subsequent research in locusts revealed that ACP promoted lipid oxidation and utilization, maintaining lipid metabolic homeostasis [23,24]. A recent study found that ACP played a role in the sperm ducts of adult spiny lobster *Panulirus ornatus* [25]. Additionally, it revealed that the ACP receptor was expressed in the reproductive system and affected the female reproduction of *Aedes aegypti*, indicating the complex sex-specific actions of the ACP signaling pathway in arthropods [26,27].

Vitellogenesis is a crucial process in the reproduction of female crustaceans, which enables the production of eggs and the development of embryos [28]. It has been shown that vitellogenesis is regulated by a complex network of factors derived from the neuroendocrine system, in which the hormones secreted by the X-organ/sinus gland (XO/SG) complex in the eyestalk ganglia play a central role [29]. For instance, vitellogenesis inhibiting hormone (VIH) is the main hormone that serves an inhibitory role in ovarian development [28]. In aquaculture, unilateral eyestalk removal has been employed to induce ovarian development and spawning in some crustaceans by alleviating the inhibitory effect of VIH, e.g., prawn [30,31]. However, this method causes nonnegligible side effects, including permanent damage, poor gamete quality, and low fertilization rate [32]. It is important to develop novel techniques that can overcome those drawbacks. Recently, GnRH-like peptides have been discovered in crustaceans, but their roles in reproduction are poorly understood. Therefore, it is intriguing to explore whether ACP is involved in regulating ovarian development in the mud crab, which raises an opportunity to explore a new approach to meet this challenge.

Mud crab (*Scylla paramamosain*) is a commercially significant crustacean in aquaculture. The gonadosomatic index (GSI) in females is usually used to assess the breeding maturation status of the animals in aquaculture, which is significant to the artificial breeding technology of *S. paramamosain*. In this study, the cDNAs encoding *Sp-ACP* and *Sp-ACPR* were first identified, and their expression profiles were subsequently investigated by reverse transcription PCR (RT-PCR) and immunofluorescence analysis. To evaluate the effect of ACP on ovarian development in *S. paramamosain*, the role of ACP in vitellogenesis was investigated by an in vitro experiment. Finally, we performed the in vivo experiments by injection of synthetic *Sp*-ACP into female *S. paramamosain* to further demonstrate its stimulatory effect on ovarian development.

## 2. Materials and Methods

### 2.1. Animals

The ovarian development of *S. paramamosain* has been classified into previtellogenic, early vitellogenic, and late vitellogenic stages, which can be distinguished by the morphological appearance and histological characteristics [33,34]. Female crabs at the three different vitellogenic stages, previtellogenic stage (body weight 65.4 ± 7.8 g, carapace width 6.2 ± 1.1 cm), early vitellogenic stage (body weight 255.9 ± 7.8 g, carapace width 6.8 ± 2.2 cm), and late vitellogenic stage (body weight 358.3 ± 11.2 g, carapace width 11.2 ± 0.8 cm), were purchased from local fish markets in Xiamen city, Fujian Province, China and then transported to the laboratory in Jimei university. The crabs were housed individually in rectangular tanks filled with seawater at a salinity of 28 ± 0.5 ppt for one week. During this period, water temperature was maintained at 27 ± 2 °C, and they were fed with clam *Ruditapes philippinarum* to satiation (approximately 10% body weight of the crab) once daily.

For tissue sampling, female crabs were placed on ice for anesthetization before tissue dissection. Tissues including eyestalk ganglia, cerebral ganglia, thoracic ganglia, ovary, hepatopancreas, heart, middle gut, gill, muscle, stomach, and the Y-organ were collected from crabs at the early vitellogenic stage for tissue distribution analysis. Meanwhile, eyestalk ganglia, cerebral ganglia, and ovary samples at the three different vitellogenic stages (*n* = 5) were also collected for *Sp-ACP* and its putative receptor expression analysis. Tissues were immediately frozen in liquid nitrogen and stored at −80 °C until total RNA extraction.

### 2.2. RNA Extraction, cDNA Synthesis, and Molecular Cloning

Total RNA was extracted from tissues using TRIzol^®^ reagent (Invitrogen, Waltham, MA, USA) according to the manufacturer’s instructions. The quantity and quality of RNA were determined by 1.5% *w/v* agarose gel electrophoresis and NanoDrop ND-1000 spectrophotometer (Thermo Fisher Scientific, Wilmington, DE, USA), and potential genomic DNA contamination was eliminated by DNase I (Takara, Shiga, Japan). The first cDNA was synthesized with the Perfect Real-Time Primer Script^®^ RT reagent Kit (Takara, Shiga, Japan). The generated cDNAs were diluted four-fold and stored at −80 °C until use.

Two fragments encoding ACP and ACPR were identified from the ovary transcriptome dataset of the mud crab *S. paramamosain*. Subsequently, they were verified by PCR with primer pairs ACP-F/R and ACPR-F/R, respectively. Primers used in this study are listed in Appendix A. The amplified DNA fragments were detected by using a 1.0% agarose gel. After being gel-purified, PCR products were ligated to pMD19-T vector for sequencing (Sangon Biotech, Shanghai, China). The opening reading frame was predicted by the ORF finder tool (https://www.ncbi.nlm.nih.gov/orffinder/, accessed on 19 March 2024), and the signal peptide was checked by Signal P software (http://www.cbs.dtu.dk/services/SignalP/, accessed on 18 July 2024). The dibasic cleavage and modification sites of the *Sp*-ACP precursor were further analyzed by NetPhos (http://www.cbs.dtu.dk/services/NetPhos/, accessed on 18 July 2024). The seven transmembrane domains and potential motifs were predicted using the webserver described in the previous study [35]. Multiple sequence alignment was conducted using Genedoc software 2.7.0 (Kar Nicholas, Pittsburgh, PA, USA). Phylogenetic trees were constructed using the neighbor-joining (NJ) method on MEGA 11 software (MEGA, Tempe, AZ, USA), with bootstrap sampling repeated 1000 times to assess the reliability of the tree.

### 2.3. Expression Profiles of Sp-ACP and Its Putative Receptor 

Tissue distribution profiles of *Sp-ACP* and *Sp-ACPR* were detected by RT-PCR with cDNAs of 11 tissues from a female *S. paramamosain* at the early vitellogenic stage. Briefly, the reaction system was in a 25 μL volume containing 12.5 μL 2X Premix Ex TaqII (Takara, Shiga, Japan), 1.0 μL cDNA, 0.5 μL each primer (10 μM), and 10.5 μL of deionized water. The PCR was performed under the following program: 94 °C for 3 min, 32 cycles of 94 °C for 30 s, 56 °C for 30 s, and 72 °C for 30 s, and a final extension at 72 °C for 10 min. The amplification of deionized water was set as the blank control, and the *β-actin* (a housekeeping gene) was concurrently amplified as an internal control. Finally, the PCR products were analyzed by 1% agarose gel electrophoresis and imaged using a UV detector (Tanon, Shanghai, China).

Real-time quantitative PCR (qPCR) was performed to profile the expression of *Sp-ACP* in eyestalk ganglia, cerebral ganglia, and ovary during ovarian development, as well as the expression of its receptor in ovary. The reaction was performed on a QuentStudio Q5 real-time PCR detection system (Applied Biosystems, Carlsbad, CA, USA) in a total volume of 20 μL, containing 2 μL diluted cDNA, 10 μL 2X PCR Master Mix with SYBR GREEN (Life Technologies, Carlsbad, CA, USA), 1 μL forward and reverse primers (1 mM), and 6 μL deionized water. The reaction was incubated for 3 min at 95 °C, followed by 40 cycles of 95 °C for 20 s, 58 °C for 30 s, and 72 °C for 30 s. Each sample was run in triplicate, and the results were normalized to the internal control gene *β-actin*. Finally, the data were calculated using 2^−ΔΔCt^ method. 

### 2.4. Localization of Sp-ACP in the Ovary by Immunofluorescence

To identify the localization of *Sp*-ACP in ovary, a custom-produced polyclonal rabbit antibody against the mature peptide of *Sp*-ACP (pQITFSRSWVPQamide) was generated (GL Biochem Ltd., Beijing, China). After prolonged immunization, a serum containing *Sp*-ACP antibody was collected, and this antibody was subsequently purified by affinity chromatography. Finally, the efficiency of antibody was determined by enzyme-linked immunosorbent assay (ELISA), and it was stored at −80 °C until use. Ovary samples were fixed with 4% paraformaldehyde solution and then embedded in paraffin, followed by being sectioned with a thickness of 7 μm. Subsequently, the samples were incubated with EDTA antigen retrieval solution (pH 8.0; Sangon Biotech, Shanghai, China) at 99 °C for 20 min and then blocked with 5% bovine serum albumin (BSA) in PBS (pH 8.0) at 25 °C for 30 min. Next, the samples were incubated with *Sp*-ACP antibodies (1:500 dilution) or preimmunized serum at 37 °C for 2 h. After being washed with phosphate buffer solution (PBS), samples were incubated with goat anti-rabbit IgG conjugated to Alexa Fluor^®^ 647 (1:500 dilution, Abcam, Cambridge, MA, USA) for 1 h. The cell nuclei were labeled by 4,6-diamidino-2-phenylindole (DAPI) at a concentration of 1 μg/mL (Beyotime, Shanghai, China). Finally, the samples were imaged by using an Olympus Ckx63 system.

### 2.5. In Vitro Effect of Sp-ACP on Sp-Vg and Sp-VgR Expression

To explore a potential function of *Sp*-ACP on vitellogenesis of *S. paramamosain*, in vitro experiments that applied biosynthesis ACP to hepatopancreas and ovary explants were performed. *Sp*-ACP (pQITFSRSWVPQamide), with a purity of 98%, was synthesized by GL Biochem., Ltd (Shanghai, China). Female crabs at the early vitellogenic stage were anesthetized on ice for 10 min, followed by sterilization in 75% ethanol for 10 min. At the end of this period, tissues of ovary and hepatopancreas were dissected and washed with crab saline containing double antibiotics, penicillin G (300 mg/mL), and streptomycin (300 mg/mL), nine times. The samples were subsequently cut into fragments of approximately 50 mg and placed in a well of 24-well culture plate with 500 μL culture medium containing double antibiotics for pre-incubation at 26 °C. There are four treatments in this experiment: without any peptide, group 1 was set as a control, while with three different concentrations (10, 100, and 1000 nM) of *Sp*-ACP into groups 2 to 4, respectively. Samples of ovarian and hepatopancreatic explants were collected post 2-, 4-, and 6-h incubation. Each sampling point has four replicates (*n* = 4). Finally, the samples were used for the detection of *Sp-Vg* (GenBank accession number: FJ812090.1) and *Sp-VgR* (GenBank accession number: KF860893.1) expressions by qPCR.

### 2.6. In Vivo Effect of Sp-ACP on Ovarian Development

Based on the result of the in vitro experiment, a stimulatory role of *Sp*-ACP in ovarian development of the mud crab was proposed. To confirm this hypothesis, both short- and long-term in vivo experiments were conducted in this study. In the short-term experiment, females at the early vitellogenic stage were randomly divided into two groups, with 7 individuals in each group (*n* = 7). Crabs were received *Sp*-ACP (15 ng/g body mass) that prepared in 100 μL crab saline, while the control group received 100 μL crab saline instead. Approximately 12 h post-injection, crabs were killed after anesthetization on ice, and hemolymph sample was collected for 17β-estradiol (E_2_) determination. In addition, samples of hepatopancreas and ovary were dissected for gene expression analysis by qPCR, as described in Section 2.3.

In the long-term experiment, female crabs at the early vitellogenic stage were randomly divided into three groups, and each group had 7 individuals (*n* = 7). Before the injection, a group of crabs was sampled and set as the pre-injection control. Crabs were injected *Sp*-ACP (15 ng/g body weight) prepared in 100 μL crab saline once every five days, while the control group received 100 μL crab saline instead. On day 16, approximately 24 h after the third injection, crabs were placed on ice for anesthetization before being killed, and hemolymph sample was collected for E_2_ measurement. Additionally, GSI measurements were calculated by function GSI= ovary weight/body weight × 100%, and samples of hepatopancreas and ovary were dissected for gene expression analysis. The histological changes in ovary were analyzed by hematoxylin and eosin staining, and the oocyte diameter was measured by Image J Version 1.54.

The concentration of E_2_ in hemolymph was measured by DetectX^®^ SERUM 17β-Estradiol Enzyme Immunoassay Kit (Arbor Assays, Ann Arbor, MI, USA). E_2_ was extracted with 5 mL diethyl ether from 1 mL hemolymph, which was subsequently applied to assay according to the manufacturer’s instructions. Each set of standards and samples was run in duplicate. Finally, the concentration of the E_2_ was calculated using Arbor Assay, an online tool (www.myassays.com/arbor-assays-estradiol-serum-eia-kit.assay, accessed on 18 July 2024).

### 2.7. Statistical Analysis

Statistical analysis was performed by SPSS 26.0 software. Levene’s test was used to assess the homogeneity of variance, and statistical significance was determined by one-way analysis of variance (ANOVA), employing Duncan’s test, in which *p* < 0.05 was judged significant, while *p* < 0.01 was judged extremely significant. All data are presented as mean ± SEM.

## 3. Results

### 3.1. Molecular Cloning of Sp-ACP and Sp-ACPR

The open reading frame (ORF) of the *Sp-ACP* sequence was 225 bp long (GenBank accession: PP885732.1), encoding a precursor with 74 amino acids (aa) in length. The precursor contains a 21-aa signal peptide, an 11-aa mature peptide (pQITFSRSWVPQamide), followed by a dibasic cleavage site (KR) and a related peptide (Figure 1). A conserved glutamine recognition site was found at the C-terminal of mature peptide, suggesting amidation occurred during the post-translational processing (Figure 1). Sequence alignment revealed that the mature ACP was highly conserved among crustaceans (Figure 2A). In addition, there was a variable pyrophosphorylation site in the N-terminal of the mature peptide, which was proline in Brachyuran crabs while replaced by alanine in shrimps and prawns (Figure 2B). The phylogenetic tree suggested that ACP, AKH, Crz were divided into three clades and all of the ACPs were fit into a same clade that between AKH and Crz clades (Figure 3).

The ORF of *Sp-ACPR* is 1569 bp long (GenBank accession: PQ590310), encoding a 522 aa protein, which contains an extracellular N-terminus, an intracellular C-terminus, seven transmembrane domains (TMs), three extracellular loops (ECLs), and three intracellular loops (ICLs). Sequence alignment revealed that *Sp*-ACPR showed an extremely high sequence similarity with other reported receptors in the TMs, which possess a DRYFAV motif, a CWTP motif, and an NPIIY motif (Figure 4). The phylogenetic tree was constructed to obtain a better insight into the evolutionary relationships among arthropod AKH, ACP, and Crz receptors. It showed that the receptors were categorized into three different clades, and all the ACP receptors were positioned within a single clade that was a sister group to the AKH/PRCH receptors clade. The AKH/RPCH and ACP receptor clades form a monophyletic cluster, which was a related group to the Crz receptors clade. Unsurprisingly, the obtained *Sp*-ACP receptor in this study was clustered closely with the well-identified ACPRs in insects, including *Tribolium castaneum*, *Ixodes scapularis*, *Anopheles gambiae*, *Bombyx mori*, and *Nasonia vitripennis* (Figure 5).

### 3.2. Expression Profiles of Sp-ACP and Sp-ACPR mRNA

RT-PCR suggested that the *Sp-ACP* was extensively expressed in the nervous tissues (eyestalk, cerebral, and thoracic ganglia), ovary, middle gut, and the Y-organ of female *S. paramamosain* (Figure 6A). In addition, *Sp-ACPR* was highly expressed in the ovary and had a low expression level in the nervous tissues and Y-organ. The expression profiles of *Sp-ACP* and *Sp-ACPR* during ovarian development were subsequently detected by qPCR. It showed that the expression of *Sp-ACP* in eyestalk ganglia was significantly increased at the late vitellogenic stage, while in the cerebral ganglia and the ovary, it was dramatically upregulated at the early and late vitellogenic stages (Figure 6B). *Sp-ACPR* showed an opposite expression pattern in the ovary to that of *Sp-ACP* during ovarian development (Figure 6C). Its expression level was high at the previtellogenic stage and gradually decreased at the early and late vitellogenic stages (*p* < 0.01 and *p* < 0.001).

### 3.3. Localization of Sp-ACP Expressing Cells in the Ovary

Profiling the spatiotemporal expression pattern of *Sp-ACP* revealed that it had a high expression in the ovary of the mud crab. Thus, we further detected its cell localization by immunofluorescence using a custom-produced antibody against *Sp*-ACP. The result showed that *Sp*-ACP was specifically localized in the follicle cells of the ovary (Figure 7).

### 3.4. In Vitro Effects of Sp-ACP on Sp-Vg and Sp-VgR Expression

The synthetic *Sp*-ACP was applied to the culture media containing ovarian and hepatopancreas explants at the early vitellogenic stage, respectively, to explore its effect on vitellogenesis. The qPCR results showed that *Sp*-ACP significantly induced the expression of *Sp-VgR* in the ovary at a final concentration of 10 nM after 4 and 6 h incubation (Figure 8D–F). On the other hand, the levels of *Sp-Vg* transcript in hepatopancreas showed no statistical difference after the addition of *Sp*-ACP (Figure 8A–C).

### 3.5. In Vivo Effects of Synthetic Sp-ACP on Ovarian Development 

It was revealed that the expression of *Sp-Vg* in the hepatopancreas and *Sp-VgR* in the ovary was significantly increased by 12 h injection of *Sp*-ACP (Figure 9A,B). Similarly, the level of hemolymph E_2_ titer was significantly increased in response to the 12 h injection of *Sp*-ACP (Figure 9C).

In addition, long-term injection of *Sp*-ACP significantly induced the expression of *Sp-Vg* (*p* < 0.05) and *Sp-VgR* (*p* < 0.01) and increased the level of hemolymph E_2_ titer and GSI of mud crabs when compared to the crab saline control group (Figure 10). Histological analysis showed that the administration of *Sp*-ACP markedly elevated the oocyte diameter and the number of yolk granules (Figure 11).

## 4. Discussion

ACP is a member of the GnRH-like superfamily in arthropods, which is structurally similar to AKH and Crz. Previous studies indicated that ACP signaling may function independently from AKH and Crz signaling pathways [21]; however, up to now, the precise role of ACP signaling in arthropods has not been well understood. In this study, we isolated an ACP and its putative receptor from the ovary and subsequently explored its possible role in the mud crab *S. paramamosain*, demonstrating that it may play a possibly stimulatory role in ovarian development.

In the mud crab, *S. paramamosain*, a sequence encoding ACP precursor was isolated from the ovary by PCR, which consisted of a signal peptide, mature peptide, and related peptide flanked by a dibasic cleavage site (KR) and an amidated signal (Figure 1). It exhibited a similar feature to the ACP precursor in insects [36]. The mature *Sp*-ACP was identified as pQITFSRSWVPQa, which had a conserved N-terminal sequence “QXTFSXXW” and an amidated C-terminal, which was essential for its bioactivity [37]. Multiple sequence alignment showed that mature ACP was highly conserved in crustaceans, with only the position of the pyrophosphorylation site being different (Figure 2). Interestingly, it was proline in brachyuran crabs, whereas it was instead by an alanine in shrimps and prawns, which could be a variable site among crustaceans. Phylogenetic analysis showed that *Sp*-ACP belonged to the ACP clade between the AKH and Crz clades in the phylogenetic tree (Figure 3), consolidating the fact that ACP was the structural intermediate to the AKH and the Crz. In addition, the ACP clade is close to the AKH branch, indicating that ACP is likely arising from duplication of the AKH gene [12,38]. 

The first ACP receptor was identified from the *A. gambiae* genomic data, which was highly related to the AKH and Crz receptors but could not be activated by AKH, Crz, or any other peptide in *A. gambiae* [21]. Subsequently, candidate ACP receptors were found in other insects and further deorphaned, such as *R. prolixus* [38] and *I. Scapularis* [39]. Recently, predicted ACP receptors have been found in the genomic dataset of several crustacean species by searching for the conserved DRYFAV motif, the CWTP motif, and the NPIIY motif [35]. In this study, the putative *Sp-ACPR* sequence was isolated, which also contained these three motifs (Figure 4). The third extracellular loop of ACPR is essential for ACP binding and receptor activation [40]. Multiple sequence alignment revealed that ACPR in crustaceans showed a high similarity in the seven transmembrane domains. The phylogenetic tree further confirmed that the receptor isolated in this study is an ortholog of other ACP receptors, which fit into the same clade of the ACP receptors in insects (Figure 5) [21,39,41]. Otherwise, a cell assay is needed to consolidate it, which can be activated by ACP in the future. In addition, given the fact that CrzR, AKHR, and ACPR are more closely related to each other, there is a possibility that they may originate from gene duplication of a common ancestor [42].

To date, the physiological role of ACP in arthropods is still unclear. Profiling the expression of ACP and its putative receptor can provide a cue to understand its possible functions in mud crab, *S. paramamosain*. In this study, *Sp-ACP* was detected in the nervous tissues, middle gut, and Y-organ and highly expressed in the ovary. This is similar to the findings in insects, *ACP* has also been observed in nervous tissues [22,43]. ACP was identified in the reproductive tissues of *P. ornatus*, suggesting it may play a possible role in the reproductive process [25]. In the mud crab *S. paramamosain*, *Sp-ACP* showed a high expression in the ovary, and the immunofluorescence result further revealed that *Sp*-ACP was specifically expressed in the follicular cells. Similarly, positive immunoreaction of GnRH-like was detected in the cytoplasm of the follicle cells of *P. monodon* [44]. This result indicates its possible role in production since follicle cells are associated with oocyte maturation, mediating the uptake of yolk proteins and involved in the secretion of related hormones [45]. *Sp-ACPR* showed a high expression in the ovary and a low expression in the nerve tissues and Y-organ, which emphasized the involvement of ACP in ovarian development (Figure 6A). The co-expression of *Sp*-*ACP* and *Sp*-*ACPR* in the ovary supported the previous findings that cell communication existed in the ovary of *S. paramamosain*, which is essential for ovarian development [46,47,48]. During ovarian development, the level of *Sp-ACPR* transcript in nervous tissues and ovary was significantly decreased (Figure 6C), indicating that *Sp*-ACP may be mainly in the regulation of ovary development during the early stage of ovarian development, whereas in the late stage, it mainly focuses on the regulation of nervous tissues. This possibility is supported by the previous study of *M. rosenbergii*, where *Mro*-ACPR-like was expressed at all stages, and its expression decreased with ovarian development, but the expression in neural tissues increased [37]. In addition, *Sp-ACPR* exhibited a weak expression in the nervous tissues, suggesting that ACP may act as a neurotransmitter and/or a neuromodulator. The presence of *Sp-ACPR* in the Y-organ, which is the source of crustacean-specific regulation of the molt cycle responding to external environmental and internal physiological signals [49], indicates a possible regulation of ecdysteroid biosynthesis and molting. It has been revealed that Crz plays an important role in molting by regulating ecdysteroid synthesis in the swimming crab *Portunus trituberculatus* [50].

The documented studies regarding ACP focused on exploring its involvement in the processes of energy storage and metabolism in insects because its high structurally similarity to AKH, which plays an important role in energy metabolism [23,24]. In the two-spotted cricket, *Gryllus bimaculatus*, ACP regulated hemolymph carbohydrate and lipid levels, possibly collaborative contribution with AKH to the maintenance of energy homeostasis [51]. However, recent evidence has revealed that ACP is also involved in the regulation of reproductive processes. For example, in the diamondback moth *Plutella xylostella*, silencing the *Px-ACPR* gene by microRNA or dsRNA resulted in pupal malformations and low spawning and hatching rates [52]. In this study, we performed in vitro and in vivo experiments to explore the function of ovarian development in the female mud crab. Our in vitro results showed that *Sp-VgR* was significantly induced by the addition of synthetic *Sp*-ACP (Figure 8). VgR is an oocyte membrane receptor, which plays an important role in ovarian development through mediating the uptake of vitellogenin by oocyte endocytosis [53]. Such a result suggested that *Sp*-ACP might promote ovarian development by inducing the expression of *Sp-VgR*. Further, the in vivo evidence consolidated this conclusion, which showed that injection of *Sp*-ACP significantly increased the expression of *Sp-Vg* and *Sp-VgR* expression (Figure 10A,B). Furthermore, oocyte diameter and the number of yolk granules in the oocytes increased after the administration of *Sp*-ACP (Figure 11), suggesting the growth of oocytes was promoted by *Sp*-ACP. A previous study revealed that Crz signaling has a similar effect, it showed that disrupting of the *CrzR* gene led to a significant decrease in *Pt-VgR* expression in the ovary of swimming crab *P. trituberculatus* [20]. Similarly, a significant increase in oocyte diameter was observed after the injection of synthetic RPCH into the mud crab *S. paramamosain* [54]. These findings enhance the significant importance of GnRH-like peptides in the ovarian development of crustaceans. In summary, our results showed that 15 ng/g might be an ideal dose of ACP for proper ovarian maturation in *S. paramamosain*, which may be applied to the artificial breeding technology of the mud crab.

In this study, the expression of *Sp-Vg* was not significantly changed by *Sp*-ACP, which is opposite the in vivo result that injection *Sp*-ACP induced the expression of *Sp-Vg*. It is a reasonable phenomenon that *Sp*-ACP did not affect gene expression in hepatopancreas directly since the putative ACP receptor is absent in this tissue. However, the in vivo evidence suggested *Sp*-ACP might regulate the *Sp-Vg* expression through the 17β-estradiol (E_2_). Past studies have shown that E_2_ was necessary for ovarian development in crustaceans. E_2_ levels in the hemolymph, ovary, and hepatopancreas were closely related to the ovarian development stage of the Chinese mitten crab, *Eriocheir sinensis* [55]. E_2_ induced *Vg* synthesis and primary vitellogenesis in the shrimp ovary [56,57]. In this study, the hemolymph E_2_ titer was significantly increased after 12 h and prolonged injection of *Sp*-ACP injection (Figure 9C and Figure 10D), suggesting that *Sp*-ACP may indirectly regulate ovarian development by inducing the expression of *Vg* in hepatopancreas through E_2_.

## 5. Conclusions

In conclusion, this study isolated sequences encoding *Sp-ACP* and its putative receptor (*Sp-ACPR*) and further assigned their stimulatory role in the ovarian development of the mud crab *S. paramamosain*. Our results suggested that *Sp*-ACP is mainly expressed by the follicular cells in the ovary and could directly induce *Sp-VgR* expression, therefore accelerating oocyte growth and ovarian development. In addition, *Sp*-ACP might stimulate *Sp-Vg* expression indirectly by increasing E_2_ titer in hemolymph and further promoting ovarian development of *S. paramamosain*. This is the first report on the stimulatory role of ACP in ovarian development in crustaceans, providing new insight into the function of ACP signaling in arthropods.

## Figures and Tables

**Figure 1 animals-14-03706-f001:**
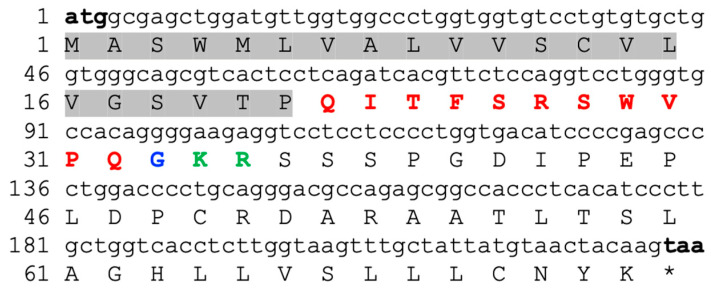
ORF of *Sp-ACP* and its deduced amino acid sequence in the mud crab *S. paramamosain*. The predicted N-terminal signal peptide is highlighted in gray background. The start and stop codons are in bold. The mature peptide of ACP is highlighted in red. The dibasic cleavage site is indicated in green, and the C-terminal amidated glycine is shown in blue. “*” represents stop codon.

**Figure 2 animals-14-03706-f002:**
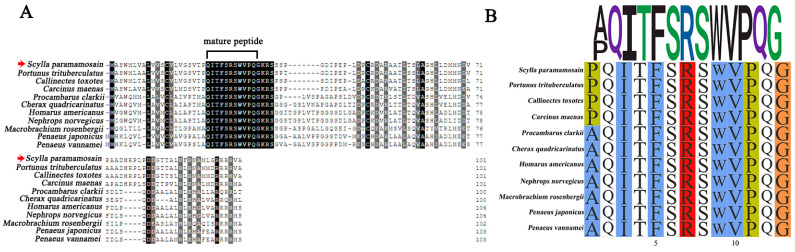
(**A**) Multiple sequence alignment of *Sp*-ACP and its homologs. The same amino acids are highlighted in black background while the similar amino acids are indicated in dark grey background. The identified ACP in *S. paramamosain* is indicated by red arrows. (**B**) Multiple sequence alignment of the mature ACP. *Portunus trituberculatus* (XP_045107636.1), *Callinectes toxotes* (QPO25130.1), *Carcinus maenas* (AVA26881.1), *Procambarus clarkii* (XP_045596871.1), *Cherax quadricarinatus* (XP_053637656.1), *Homarus americanus* (XP_042242726.1), *Nephrops norvegicus* (QBX89024.1), *Macrobrachium rosenbergii* (ANT96502.1), *Penaeus japonicus* (XP_042893058.1), *Penaeus vannamei* (XP_027229238.1).

**Figure 3 animals-14-03706-f003:**
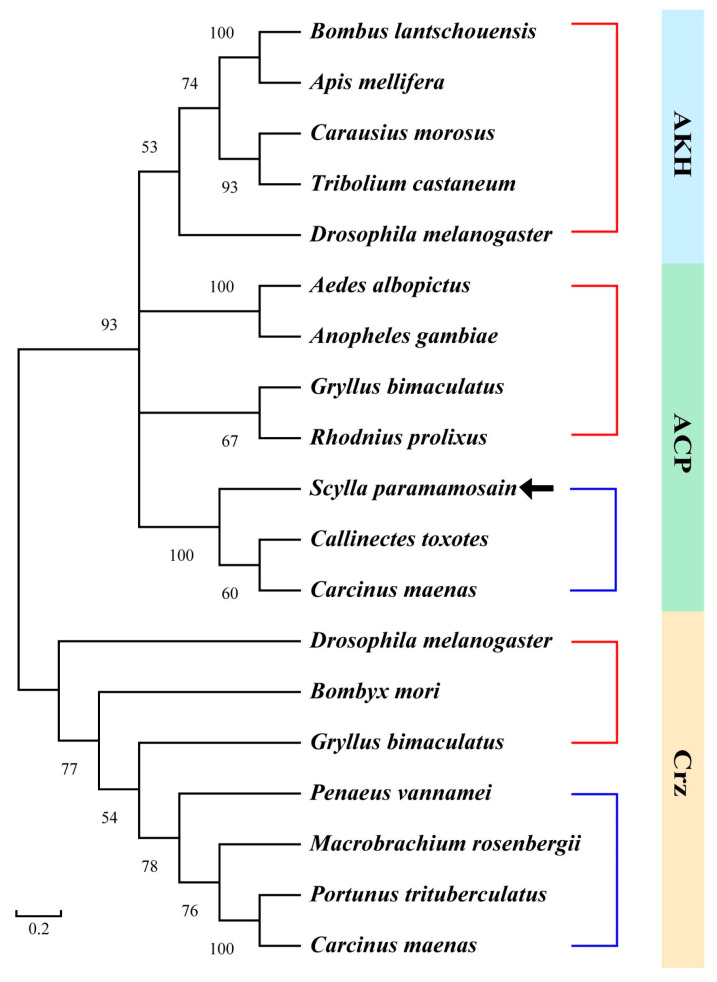
Phylogenetic analysis of AKH, ACP, and Crz in arthropods. Red and blue branches represent insects and crustaceans, respectively. The GenBank accession numbers of proteins used for phylogenic analysis are as follows: AKH: *Bombus lantschouensis* (QGN75352.1), *Apis mellifera* (AEW68342.1), *Carausius morosus* (UES72857.1), *Tribolium castaneum* (NP_001107818.1), *Drosophila melanogaster* (NP_523918.1); ACP: *Aedes albopictus* (XP_029718603.1), *Anopheles gambiae* (XP_563757.1), *Gryllus bimaculatus* (BDY34174.1), *Rhodnius prolixus* (AKO62855.1), *Callinectes toxotes* (QPO25130.1), *Carcinus maenas* (AVA26881.1); Crz: *Drosophila melanogaster* (NP_524350.1), *Bombyx mori* (BAC66443.1), *Gryllus bimaculatus* (BDY34175.1), *Penaeus vannamei* (WPR17731.1), *Macrobrachium rosenbergii* (ALA65535.1), *Portunus trituberculatu* (UZH25338.1), *Carcinus maenas* (AVA26882.1).

**Figure 4 animals-14-03706-f004:**
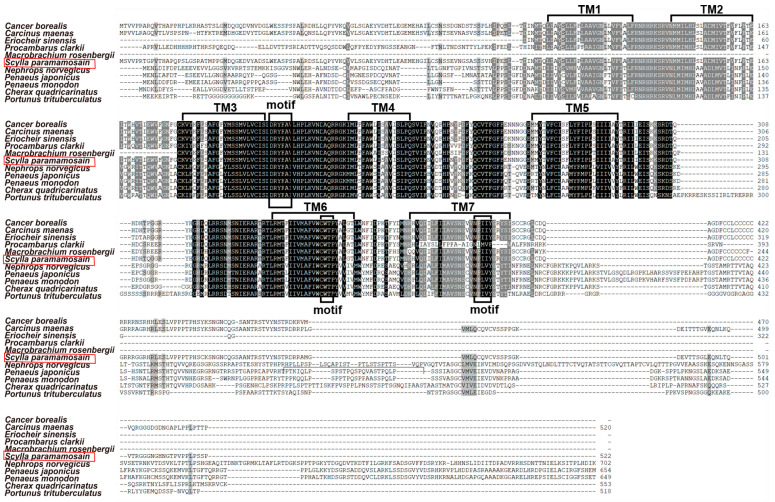
Multiple sequence alignment of deduced amino acid sequences from *Sp*-ACPR and its homologs. *Eriocheir sinensis* (XP_050725553.1), *Procambarus clarkii* (XP_069178490.1), *Macrobrachium rosenbergii* (XP_064110518.1), *Penaeus japonicus* (XP_042883232.1), *Penaeus monodon* (XP_037774021.1), *Cherax quadricarinatus* (XP_053652064.1), *Portunus trituberculatus* (XP_045136970.1). The sequence of ACPR in *Cancer borealis*, *Carcinus maenas*, and *Nephrops norvegicus* was obtained from [35]. TM: transmembrane domain. The same amino acids are highlighted in black background, and the similar amino acids are indicated in dark grey background. The identified ACPR in *S. paramamosain* is indicated by red boxes.

**Figure 5 animals-14-03706-f005:**
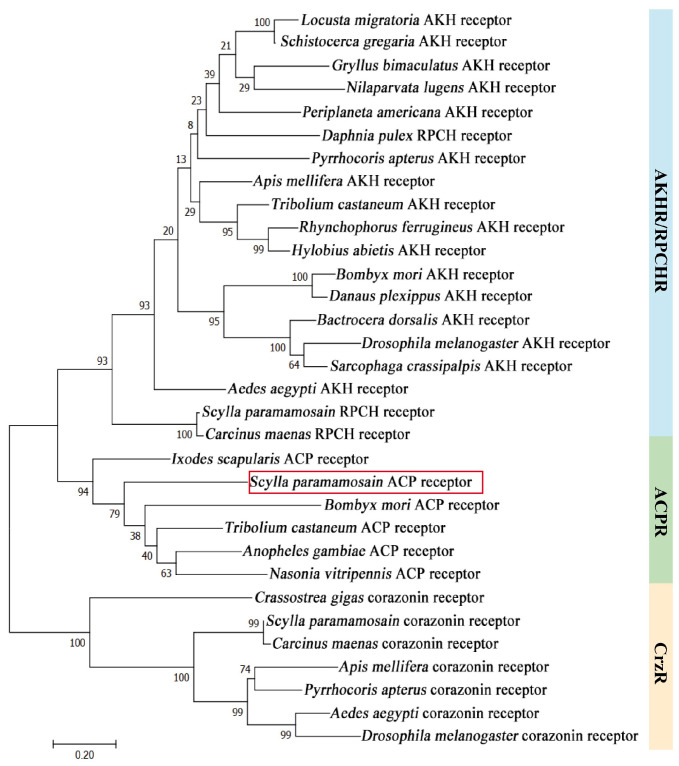
Phylogenetic analysis of AKHR, RPCHR, ACPR, and CrzR sequences in arthropods. The GenBank accession numbers of proteins used for phylogenic analysis are as follows: AKHR: *Locusta migratoria* (ANW09575.1), *Schistocerca gregaria* (AVG47955.1), *Gryllus bimaculatus* (ADZ17179.1), *Nilaparvata lugens* (AZP54622.1), *Periplaneta americana* (ABB20590.1), *Daphnia pulex* (ARD09217.1), *Pyrrhocoris apterus* (ARV86499.1), *Apis mellifera* (NP_001035354.1), *Tribolium castaneum* (NP_001076809.1), *Rhynchophorus ferrugineus* (QGA72493.1), *Hylobius abietis* (AVI00624.1), *Bombyx mori* (NP_001037049.1), *Danaus plexippus* (OWR46881.1), *Bactrocera dorsalis* (AQX83416.1), *Drosophila melanogaster* (AAS64647.1), *Sarcophaga crassipalpis* (AOC38019.1), *Aedes aegypti* (CAY77164.1); RPCHR: *Carcinus maenas* (AVA26880.1). ACPR: *Ixodes scapularis* (AHB51764.1), *Bombyx mori* (ACT79362.1), *Tribolium castaneum* (NP_001280549.1), *Anopheles gambiae* (ABX52399.1), *Nasonia vitripennis* (NP_001164571.1). CrzR: *Crassostrea gigas* (AKA95277.1), *Carcinus maenas* (AVA26879.1), *Apis mellifera* (NP_001137393.1), *Pyrrhocoris apterus* (ARV86500.1), *Apis mellifera* (NP_001137393.1), *Aedes aegypti* (AVI09462.1), *Drosophila melanogaster* (AAM21341.1). The identified ACPR in *S. paramamosain* is indicated by red box.

**Figure 6 animals-14-03706-f006:**
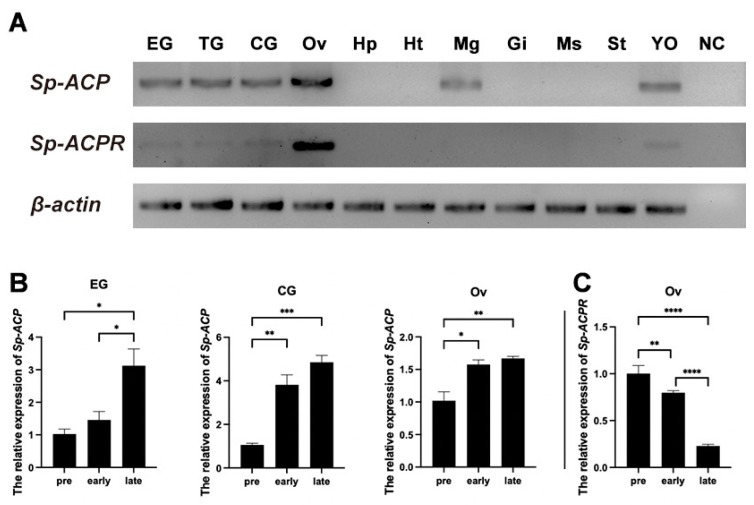
Expression profiles of *Sp-ACP* and *Sp-ACPR* transcripts in female *S. paramamosain*. (**A**) Tissue distribution of *Sp-ACP* and *Sp-ACPR* in a female mud crab at the early vitellogenic stage. EG, eyestalk ganglia; TG, thoracic ganglia; CG, cerebral ganglia; Ov, ovary; Hp, hepatopancreas; Ht, heart; Mg, middle gut; Gi, gill; Ms, muscle; St, stomach; YO, Y-organ; NC, negative control (amplification of water). (**B**) Expression profile of *Sp-ACP* in the eyestalk and cerebral ganglia, and the ovary during ovarian development. (**C**) Expression profile of *Sp-ACPR* in the ovary during ovarian development. Data are shown as means ± SEM (*n* = 5). Asterisks (* *p* < 0.05, ** *p* < 0.01, *** *p* < 0.001, **** *p* < 0.0001) indicate significant difference.

**Figure 7 animals-14-03706-f007:**
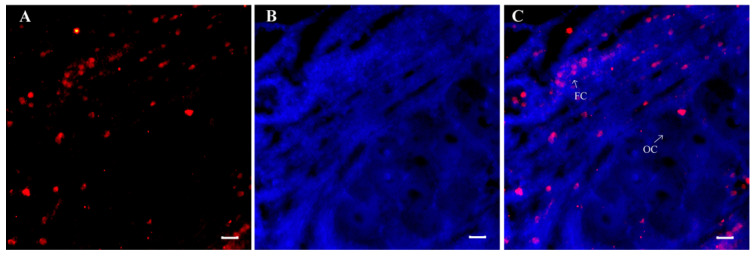
Localization of Sp-ACP expressing cells in the ovary of *S. paramamosain* by immunofluorescence. (**A**) Immunoreactive cells of *Sp*-ACP in follicle cells (red); (**B**) cell nuclei labeled by DAPI (blue); and (**C**) merge. The scale bar presented 20 μm. FC: follicular cell, OC: oocyte.

**Figure 8 animals-14-03706-f008:**
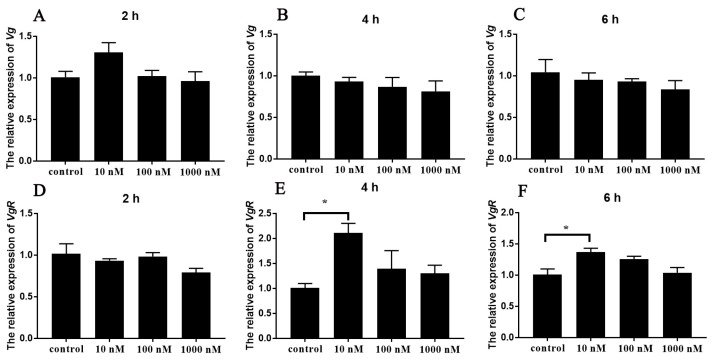
In vitro effects of *Sp*-ACP on the expression of *Sp-Vg* in the hepatopancreas (**A**–**C**) and *Sp-VgR* in the ovary (**D**–**F**). Data are shown as mean ± SEM (*n* = 4). “*” indicates a significant difference from the control (*p* < 0.05).

**Figure 9 animals-14-03706-f009:**
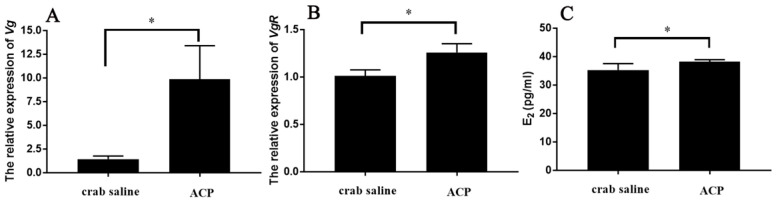
The changes in the levels of *Sp-Vg* in the hepatopancreas (**A**), *Sp-VgR* in the ovary (**B**), and hemolymph E_2_ titer (**C**) in response to 12 h injection of *Sp-ACP*. Data are shown as mean ± SEM (*n* = 7). “*” indicates a significant difference from the control (*p* < 0.05).

**Figure 10 animals-14-03706-f010:**
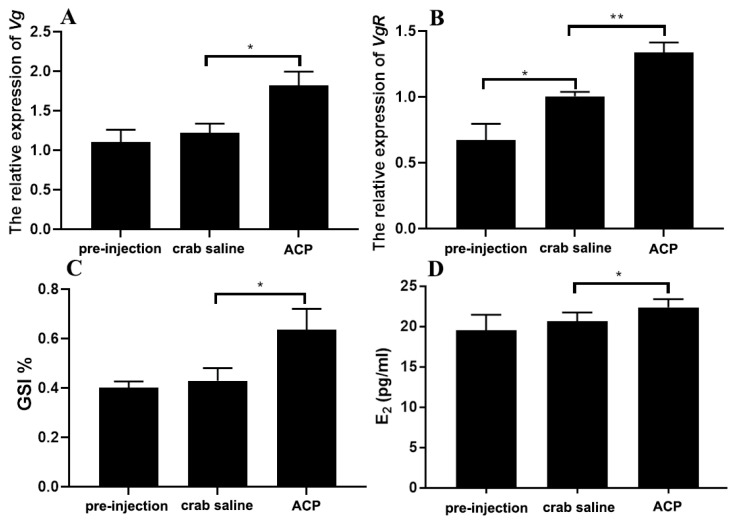
Effects of prolonged *Sp*-ACP injection on ovarian development. (**A**,**B**): Relative expression of *Sp-Vg* (**A**) in the hepatopancreas, *Sp-VgR* (**B**) in the ovary in response to prolonged injection of *Sp*-ACP. (**C**,**D**): Effects of prolonged *Sp*-ACP injection on GSI (**C**) and hemolymph E_2_ titer (**D**). Data are shown as means ± SEM (*n* = 7). Asterisks (* *p* < 0.05 and ** *p* < 0.01) on the error bar indicate significant differences from the control.

**Figure 11 animals-14-03706-f011:**
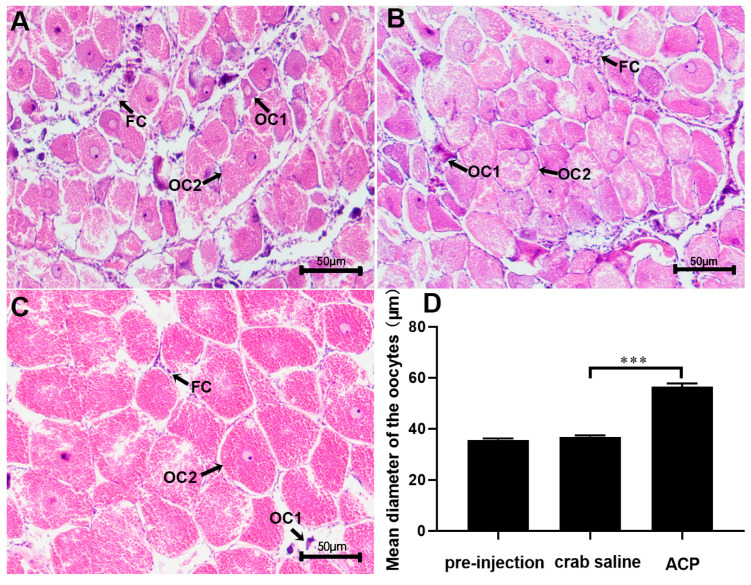
Histological changes in ovary in response to prolonged injection of *Sp*-ACP. (**A**) pre-injection control; (**B**) crab saline control; (**C**) *Sp*-ACP treatment; and (**D**) Effect of prolonged injection of *Sp*-ACP on oocyte diameter. “***” indicates a significant difference from crab saline control group (*p* < 0.001).

## Data Availability

The data presented in this study are available on request from the corresponding author.

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
