# Peer review of "Evaluation of the Effect of Adipokinetic Hormone/Corazonin-Related Peptide (ACP) on Ovarian Development in the Mud Crab, Scylla paramamosain"

_animals, 2024, doi:10.3390/ani14243706_

Round 1
Reviewer 1 Report
Comments and Suggestions for Authors
In crustaceans, the sutdies of neuroendocrinology mainly focus on the CHH peptides in the eyestalk. This MS investagates the role of the adipokinetic hormone/corazonin-related peptide (ACP) and its putative receptor in ovarian development in the mud crab Scylla paramamosain. To the best of my knowledge, the functions of ACP remains unclear in crustaceans. The author isolated the peptide ACP and its receptor by molecular cloning, and analysed the expression profiles of these two genes. They further demonstrated that ACP signaling promoted the mRNA expression of Vg in the hepatopancreas, level of estradiol in the hemolymph, and VgR expression and oocyte growth in the ovary. These results indicated that ACP signaling pathway is important in governing ovarian development in S. paramamosain, which contributes to the understanding of reproductive biology and neuroendocrinology in crustaceans.The manuscript is well-organized and well-written, with the methodology and other sections appearing to be fine. I think it can be accepted for publication after minor revisions. The suggested modifications are as follows.
1) Line 87, Mud crab (S. paramamosain), the full name should be given here.
2) Line 193, Each sample point has four replicates (n = 5). How many replicates, four replicates or five replicates in the experiment?
3) In the Figure 2A and Figure 4, the font size is too small, they needs to be enlarged.
4) Line 318, The result of heart was not provided in Figure 6. And I found there are two TG markers in this figure. Please provide the correct results.
5) The results showed that ACP was highly expressed in the nervous organs and ovary, meanwhile the ACP receptor is expressed the ovary. It is an interesting discovery. I suggested the authors should discuss the possible pathway of ACP by neurocrine or autocrine/paracrine in this crab species.
6) The references need further modification, such as the reference number 19, 20, 24, 26, 29, 34, 35.
Reviewer 2 Report
Comments and Suggestions for Authors
Review of the manuscript (animals-3358910) "evaluation of the effect of adipokinetic hormone/corazonin-related peptide (ACP) on ovarian development in the mud crab, Scylla paramamosain".
Adipokinetic hormone/corazonin-related peptide (ACP) is a GnRH structurally related peptide in arthropods, but its function is poorly understood in crustaceans. In this study, Tan et al. explored the role of ACP and its putative receptor in ovarian development in a commercially important crab, Scylla paramamosain. The results showed that Sp-ACP was highly expressed in the ovary, specifically in the follicle cells, and assigned its important role in ovarian development. Through in vivo and in vitro experiments, the authors demonstrated that ACP signaling enhanced vitellogenin (Vg) expression in the hepatopancreas, which is likely mediated by estradiol, and promoted vitellogenin uptake by oocytes through inducing Vg receptor expression. This research provides new insights into the role of ACP signaling in regulating ovarian development in crustaceans, contributing to the understanding of reproductive biology in aquaculture species.
In general, methods used are necessary, data are enough to support the main conclusion, and the manuscript is well-written and well-structured. Therefore, the manuscript can be accepted for publication after minor revisions.
1) Line 77, “bilateral eyestalk ablation”, since the unilateral eyestalk removal is the most common procedure, it is recommended to delete the word bilateral.
2) Line 87, “Mud crab (S. paramamosain)”, when a scientific name for a species appears the first time in the text, its full name should be given.
3) Line 93, delete the word “potential”.
4) Line 108, it is felled, or filled?
5) Line 258, “insect and crustacean”, both of them should be written in the plural form.
6) Line 281, Figure 4 is illegible and needs to be enlarged.
7) Line 307, the word profile should be written in the plural form.
8) Line 325-326, this sentence needs to be rewritten.
9) Line 354, change were to are
10) Line 357, prolonged injection?
11) Line 447, add the common name before Plutella xylostella.
Reviewer 3 Report
Comments and Suggestions for Authors
This research deals with reproductive physiology of an aquaculture species of crab- Scylla paramamosain. It is important work to elucidate the basic neuroendocrine biology of the species, with potential benefit of replacing the eyestalk ablation technique currently used in production of crustaceans especially in shrimp farming.
1. The structure of the experimental work is well thought through. It includes in-vitro and in-vivo work to test the hypothesis of whether Sp-ACP and its receptor is part of the breeding physiology of the crab.
2. Methods used for the cloning, identification and measuring the expression of the neuropeptide are well described and seem adequate.
3. The results of gene expression are measured quantitatively and shown by the immunofluorescence work, also correlated by histopathology to demonstrate the end-target result of ovarian development in the crabb species, using live animals.
4. The wprk could be improved by answering the several technical questions posed on the dose-response of ACP used, the histology needs scale bars on all figures, and discussion on what is the optimal dose of ACP that could result in a practical breeding outcome for these crab species.
5. There are several English grammar revisions to do.
6. Overall, the work is well designed, executed and presented. With some minor revisions, it can be fit for publication.

Minor English grammar revision required.
